# Fermentation of Habanero Pepper by Two Lactic Acid Bacteria and Its Effect on the Production of Volatile Compounds

Diego López-Salas [1], Julio Enrique Oney-Montalvo [1], Emmanuel Ramírez-Rivera [2], Manuel Octavio Ramírez-Sucre [1] and Ingrid Mayanin Rodríguez-Buenfil [1,*]

[1] Centro de Investigación y Asistencia en Tecnología y Diseño del Estado de Jalisco A.C. Sede Sureste, Tablaje Catastral 31264 km. 5.5 Carretera Sierra Papacal-Chuburna Puerto, Parque Científico Tecnológico de Yucatán, Mérida 97302, Mexico; dilopez_al@ciatej.edu.mx (D.L.-S.); juoney_al@ciatej.edu.mx (J.E.O.-M.); oramirez@ciatej.mx (M.O.R.-S.)

[2] Tecnológico Nacional de México/Tecnológico Superior de Zongolica, Departamento de Innovación Agrícola Sustentable Km. 4 Carretera S/N, Tepetlitlanapa, Zongolica 95005, Mexico; oax2010@hotmail.com

[*] Correspondence: irodriguez@ciatej.mx

**Abstract:** *Lactiplantibacillus plantarum* is a lactic acid bacterium that grows in different environments; this ability arises due to the variability within the species, which may be influenced by their origin. On the other hand, habanero pepper (*Capsicum chinense*) from Yucatan, Mexico, is characterized by its unique sensory properties such as aroma and pungency and has an annual production of more than 5000 t in the Yucatan Peninsula. Thus, the purpose of this study was to compare *L. plantarum* from different isolation sources during habanero pepper fermentation. A $2^3$ factorial design was made for the evaluation of the effect of two cultures a commercial (COM) and a wild (WIL) strain, in a habanero pepper puree medium (HPP); ripe and unripe peppers and different proportions of habanero pepper puree (40:60 or 60:40, HPP:water, $w/w$) were used to obtain the kinetic parameters of growth, lactic acid production, and volatile composition. The highest growth and lactic acid production were achieved in the 60:40 HPP:water, while WIL presented the major production of lactic acid. Characteristic volatiles in WIL fermentation were 2,3- butanedione, whereas in COM fermentation, they were limonene, cis-3-hexenyl hexanoate, and 1-hexanol. The association between COM and 1-hexanol was confirmed with principal component analysis (PCA).

**Keywords:** *Lactobacillus plantarum*; *Capsicum chinense*; volatile compounds; kinetic parameters; lactic acid

## 1. Introduction

Lactic acid fermentation of fresh foods such as meat and vegetables is a popular, sustainable, and effective process for the preservation and improvement of their sensory, nutritional, and functional attributes, as well as for increasing their shelf life [1,2]. The main representatives of the fermentative process are lactic acid bacteria, which are the most used microorganisms in the food industry [3], but they are also used for human health as probiotics [4].

Lactobacilli stand out among lactic acid bacteria for their genetic diversity and versatility in carbohydrate consumption [5]; furthermore, some species are able to perform genomic specialization to adapt to different environments, for instance, the improved development in the digestive tract of vertebrates made by *Limosilactobacillus reuteri* or the dairy products specialization through genomic decay by *Lacticaseibacillus paracasei* [6]. *Lactiplantibacillus plantarum* is another species that can be found in a variety of environments—from vegetables to the human body, thus making its genome one of the largest of all lactic acid bacteria [5]. However, the exact origin of its variability, as well as the determination of the relationship between the source of isolation and its phenotypic expression, is not well known [6].

Recently, *L. plantarum* has been successfully used in the fermentation of eggplants, carrots, pineapples, tomatoes, and some species of the *Capsicum* genus [7]. Fermentations of different types of pepper have been performed in previous studies, for instance, the spontaneous fermentation of chili pepper (*Capsicum frutescens*) and of guajillo chili (*Capsicum annum*) by three types of yeast. In both cases, fermentation modified the sensory properties of the vegetables; however, their volatile profiles differed immensely due to the distinctness of the microorganisms used, finding only propanoic acid as a common product in both cases [2,8].

Among the peppers of the *Capsicum* genus, *Capsicum chinense* is considered one of the more representative [9]. It is commonly named habanero pepper and is regarded as an important crop from Yucatan, Mexico, for (1) its high production (5049 t as of 2020) and (2) its appellation of origin, conferred in 2010. The pungency of the pepper can reach up to 350,000 Scoville heat units and its flavor and aroma are also well known [10,11]. The aroma is related to its volatile composition, which varies through maturity; unripe habanero has been reported to have a higher concentration of aldehydes, while ripe it achieves a higher presence of esters, alcohols, and terpenoids [11–14]. The use of *Capsicum chinense* as a raw material during fermentation can provide a food product of high added value and with unique organoleptic characteristics that differentiate it from other fermented products made from *Capsicum*.

In view of the above, the aim of this article was to compare two different strains of *Lactiplantibacillus plantarum*—one isolated from Yucatan habanero pepper and one for commercial use in the fermentation of habanero pepper—by evaluating kinetic parameters (growth and lactic acid production), as well as the volatile composition. The knowledge generated from the present research can be useful to develop different ways of preserving and formulating fermented habanero pepper products from Yucatan supported by its appellation of origin.

## 2. Materials and Methods

### 2.1. Habanero Pepper Obtention and Processing

Habanero pepper was obtained from a local distributor (Sabor del Mayab, Yucatán, México) in two maturity stages—unripe (green color) and ripe (orange color). Their peduncles were removed and then rinsed with tap water and disinfected with 1.75 mL L$^{-1}$ ionized silver 0.082% (*w/v*) (Microdyn®, Azcapotzalco, México) for 10 min. Finally, the peppers were blended to make a puree and preserved at −18 °C until further use.

### 2.2. Microorganisms and Growth Conditions

The strains *L. plantarum* LDL (code ECGC 13110402, SACCO, Cadorago, Italy) and *L. plantarum* YFPB1BMX previously isolated from habanero pepper at Centro de Investigación y Asistencia en Tecnología y Diseño del Estado de Jalisco, A. C. (GenBank: FJ538586.1) were used.

First, an inoculum of 10$^7$ cells·mL$^{-1}$ for each strain was incubated at 40 °C, 10 h in De Man Rogosa Sharpe (MRS) broth (DIFCO$^{TM}$, Le Pont de Claix, France); then, the fermented medium was used to inoculate another flask of MRS at a 10% (*v/v*) ratio, which was incubated at the same temperature for 6 h. The biomass was centrifuged (4 °C, 20 min, 4700 rpm in a MEGAFUGE 40R, Thermo Fisher Scientific, Bremen, Germany), followed by two rounds of a saline solution (NaCl 0.85% (*w/v*)) wash to obtain the microorganism pellet.

### 2.3. Habanero Pepper Fermentation

The pellet obtained in Section 2.2 was resuspended in habanero pepper puree medium and incubated at 40 °C for 146 h, sampling was performed every 2 h for 14 h and then every hour from 12 to 18 h. A 2$^3$ factorial design with a duplicate was conducted (Table 1) with the following kinetic parameters: growth rate (μ), maximum biomass production (ΔX), lactic acid production rate (Q$_p$), maximum lactic acid production (ΔP), maximum

productivity ($P_{max}$), and each volatile compound, as response factors. Equations for μ, ΔX, $Q_p$, ΔP, and $P_{max}$ are listed as follows:

$$dX/dt = μ \cdot t \tag{1}$$

$$ΔX = X_f − X_o \tag{2}$$

$$dP/dt = Q_p \cdot t \tag{3}$$

$$ΔP = P_f − P_o \tag{4}$$

$$P_{max} = ΔP/t_{max} \tag{5}$$

where X is the concentration of biomass expressed in dry weight (g $L^{−1}$); t is the fermentation time (h); $X_f$ is the highest concentration of biomass at a specific time (g $L^{−1}$); $X_o$ is the initial concentration of biomass (g $L^{−1}$); P is the concentration of lactic acid (g $L^{−1}$); $P_f$ is the highest concentration of lactic acid (g $L^{−1}$); $P_o$ is the initial concentration of lactic acid (g $L^{−1}$); $t_{max}$ is the maximum lactic acid production time.

**Table 1.** Design of experiments for the fermentation of habanero pepper.

| Experiment No. | Codified Variable | | | Real Variable | | |
|---|---|---|---|---|---|---|
| | **A** | **B** | **C** | **A: Strain** | **B: Proportion Puree–Water \*** | **C: Maturity** |
| 1 | − | − | − | COM | 40% | Unripe |
| 2 | + | − | − | WIL | 40% | Unripe |
| 3 | − | + | − | COM | 60% | Unripe |
| 4 | + | + | − | WIL | 60% | Unripe |
| 5 | − | − | + | COM | 40% | Ripe |
| 6 | + | − | + | WIL | 40% | Ripe |
| 7 | − | + | + | COM | 60% | Ripe |
| 8 | + | + | + | WIL | 60% | Ripe |

Note. \* distilled water. Abbreviations: COM, commercial *L. plantarum*; WIL, wild *L. plantarum*.

*2.4. Measurement of Biomass and Lactic Acid*

2.4.1. Biomass Measurement

Biomass was measured by direct microscopic count, to finally obtain the dry weight from a correlation curve. This was made with samples of commercial (COM) and wild (WIL) *L. plantarum* in MRS at concentrations of 5 and 7 × $10^9$ cells $mL^{−1}$, respectively (Figure A1, Appendix A). Different dilutions with distilled water were made for both COM and WIL, with the lowest dilution being 10% of the maximum concentration. They were centrifuged (4 °C, 20 min, 4700 rpm in a Megafuge 40R) and washed two times with distilled water. The resulting pellet was resuspended in 5 mL of distilled water and dried for 16 h at 60 °C. The dry pellet weight was measured on an analytic scale (Ohaus Explorer PA224C) at constant weight [15].

2.4.2. Determination of Lactic Acid

Lactic acid production was determined using the spectrophotometric assay developed by Borshchevskaya et al., 2016 [16], which consists of the addition of 50 μL of a lactic acid containing the sample to 2 mL of a solution of $FeCl_3$ 0.2% (*w/v*); then, the sample is homogenized, and finally, optical density is measured at 390 nm. Fermented samples were analyzed by this method, and readings of the optical density at 390 nm were made using a UV–Visible Spectrophotometer (6715 series, Jenway, Staffordshire, United Kingdom). The results were compared with a calibration curve of a standard lactic acid (≥85%, aqua solutions) from 1 to 10 g $L^{−1}$ [16].

### 2.5. Extraction of Volatile Compounds

Samples of 50 mL of the 146 h fermentation were distilled with water as a carrier solvent for one hour after it reached its boiling point. The distillate obtained was extracted with 5 mL of dichloromethane and separated by liquid–liquid extraction, recovering the dichloromethane. Finally, the samples were concentrated at 1 mL and analyzed via gas chromatography [12,17].

### 2.6. Analysis of Volatiles Compounds by Gas Chromatography

A Thermo Scientific Trace GC Ultra Gas Chromatograph equipped with a flame ionization detector was used to determine the volatile compounds. The column was a ZB-WAX plus polyethylene glycol column (60 m × 0.25 mm i.d., 0.25 μm thickness). The injection was of 2 μL at 250 °C on splitless mode. Oven temperature stayed at 40 °C for 4 min, then increased to 150 °C at a 3 °C min$^{-1}$ rate, followed by another increase to 240 °C at a 6 °C min$^{-1}$ rate; this last temperature was maintained for 2 min. Nitrogen was used as a carrier at 1 mL min$^{-1}$ [12,17].

Volatile compounds were identified by comparing the retention times with those of the evaluated standards: 2,3-buthanedione (purity ≥ 97%), limonene (purity ≥ 97%), isoamyl isobutyrate (purity ≥ 98%), trans-2-hexen-al (purity ≥ 98%), 1-hexanol (purity ≥ 99%), hexyl-3-methyl butanoate (purity ≥ 99%), 3,3-dimethyl-1-hexanol (purity ≥ 97%), linalool (purity ≥ 95%), and cis-3-hexenyl hexanoate (purity ≥ 98%). All standards were obtained from Sigma-Aldrich® (Toluca, Mexico). They were later quantified with a calibration curve from 80 to 670 μg mL$^{-1}$ (Figure A2, Table A1, Appendix B).

### 2.7. Statistical Analysis

Response variables were compared using a multifactor ANOVA and minimum significant differences (MSDs) with a 95% confidence level in the Statgraphics Centurion XVI software (Statgraphics Technologies Inc., the Plains, USA). A principal component analysis (PCA) with k means clustering was also performed for volatile compounds and experimental factors with XLSTAT 2021.2.2 (Addison, Paris, France) and with R 4.0.3 (The R Foundation for Statistical Computing, Vienna, Austria).

## 3. Results

### 3.1. Biomass and Lactic Acid Production during Fermentation

Figure 1a shows biomass production during 26 h of fermentation time. Maximum growth was achieved between 2 to 6 h of fermentation, and this rate remained stable until the end of the fermentation (146 h). Adaptation time seemed minimum in most cases, except for fermentation with commercial *L. plantarum* (COM) in unripe HPP at a 40% (*w/v*) proportion, which presented a lag phase of approximately 2 h. Exponential growth lasted 4 to 6 h, except for the fermentation of ripe HPP at 40% (*w/v*) fermented by wild *L. plantarum* (WIL), which took only 2 h. Maximum biomass production was observed in ripe HPP with a 60% (*w/v*) proportion, and COM apparently developed the highest biomass production at the stationary phase. This strain, however, seemed to have the lowest concentration at the stationary phase when the HPP was unripe and in a lesser proportion. Lactic acid production over time is presented in Figure 1b for a period of 50 h from the 146 h fermentation; it can be observed that production of this acid started immediately, reaching the highest peak at 26 h and maintaining this rate of production until the end of the fermentation (146 h). In all the conditions tested, WIL had a higher concentration of lactic acid production over time than COM, reaching its peak with a 60% (*w/v*) ripe HPP medium. Lactic acid production with 40% ripe HPP and 60% unripe HPP presented very slight differences regardless of the strain.

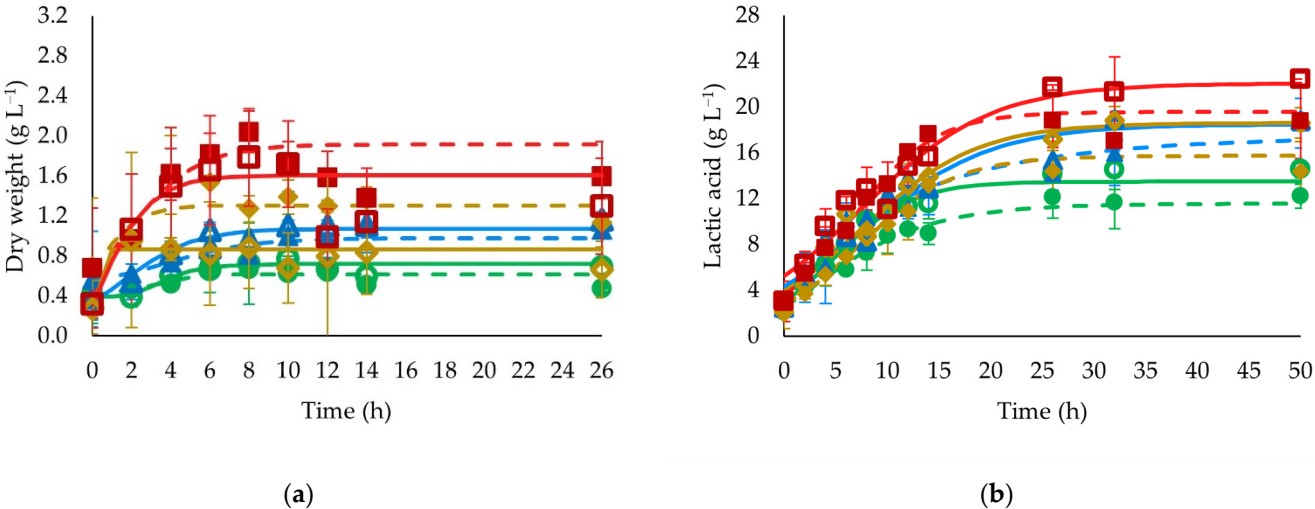

**Figure 1.** Mean biomass and lactic acid production during habanero pepper fermentation: (**a**) biomass expressed as dry weight against time; (**b**) lactic acid against time. Symbology for %HPP–water (*w/v*), ripeness, bacteria: ● 40%, unripe, COM; ○ 40%, unripe, WIL; ▲ 60%, unripe, COM; △ 60%, unripe, WIL; ◆ 40%, ripe, COM; ◇ 40%, ripe, WIL; ■ 60%, ripe, COM; □ 60%, unripe, WIL. Abbreviations: HPP, habanero pepper puree, COM, commercial *L. plantarum*; WIL, wild *L. plantarum.*

### 3.2. Evaluation of Kinetic Parameters

Table 2 reflects the values obtained for each kinetic parameter. Biomass production is represented by $\Delta X$ and $\mu$, the quantity of production, and the rate, respectively, while lactic acid production is denoted by $\Delta P$, $Q_p$, $P_{max}$, the amount of lactic acid generated, and the rate, respectively. WIL presented the highest values overall on a ripe HPP 60% (*w/v*) medium, which makes this combination the best combination to produce biomass and lactic acid. By contrast, the lowest values were obtained from COM on an unripe HPP 40% (*w/v*) medium. There were no differences between lactic acid production on a 40% (*w/v*) ripe HPP medium and a 60% (*w/v*) unripe HPP medium regardless of the strain.

**Table 2.** Kinetic parameters in the fermentation of habanero pepper.

| Sample | Strain | Proportion Puree–Water | Maturity | $\mu$ (h$^{-1}$) | $\Delta X$ (g L$^{-1}$) | $Q_p$ (g L$^{-1}$h$^{-1}$) | $\Delta P$ (g L$^{-1}$) | $P_{max}$ (g L$^{-1}$h$^{-1}$) |
|---|---|---|---|---|---|---|---|---|
| 1 | COM | 40% | Unripe | 0.0888 ± 0.0288 [a] | 0.30 ± 0.17 [a] | 0.6298 ± 0.1746 [a] | 10.36 ± 1.60 [a] | 0.39 ± 0.08 [a] |
| 2 | WIL | 40% | Unripe | 0.0839 ± 0.0306 [a] | 0.51 ± 0.41 [b] | 0.8804 ± 0.2652 [a] | 12.50 ± 1.24 [a] | 0.44 ± 0.01 [b] |
| 3 | COM | 60% | Unripe | 0.0665 ± 0.0313 [a] | 0.60 ± 0.29 [b] | 1.1904 ± 0.0977 [b] | 15.57 ± 2.31 [b] | 0.31 ± 0.05 [a] |
| 4 | WIL | 60% | Unripe | 0.1151 ± 0.0241 [a] | 0.79 ± 0.02 [b] | 0.8235 ± 0.1815 [a] | 16.40 ± 0.09 [b] | 0.51 ± 0.00 [c] |
| 5 | COM | 40% | Ripe | 0.2275 ± 0.0713 [b] | 0.91 ± 0.29 [c] | 0.8413 ± 0.0635 [a] | 12.71 ± 2.68 [a] | 0.46 ± 0.06 [b] |
| 6 | WIL | 40% | Ripe | 0.2244 ± 0.0594 [b] | 0.75 ± 0.30 [b] | 1.3428 ± 0.3152 [c] | 16.69 ± 0.25 [b] | 0.52 ± 0.01 [c] |
| 7 | COM | 60% | Ripe | 0.1967 ± 0.0872 [b] | 1.36 ± 0.40 [c] | 1.0952 ± 0.0568 [b] | 15.90 ± 1.61 [b] | 0.61 ± 0.06 [d] |
| 8 | WIL | 60% | Ripe | 0.3686 ± 0.0764 [c] | 1.96 ± 0.19 [d] | 1.4257 ± 0.2640 [c] | 20.14 ± 0.54 [c] | 0.72 ± 0.06 [e] |

Note. Data are expressed as means. Values in the same column that do not share a lower-case letter (a–e) are statistically different. Abbreviations: COM, commercial *L. plantarum*; WIL, wild *L. plantarum*; $\mu$, growth rate; $\Delta X$, maximum biomass production, $Q_p$, lactic acid production rate; $\Delta P$, maximum lactic acid production; $P_{max}$, maximum productivity.

*p* values obtained using a multifactorial ANOVA of the factors and their interactions were evaluated, to determine which had a significant effect on each kinetic parameter (Table 3). Maturity had a significant effect on all kinetic parameters, followed by the proportion of HPP–water, which had influence over three parameters ($\Delta X$, $\Delta P$, and $P_{max}$); by contrast, the strain factor had the least effect—only on two ($\Delta P$ and $P_{max}$). There were significant interaction effects between strain and maturity in relation to $Q_p$, and between proportion and maturity in relation to $P_{max}$.

**Table 3.** *p* values for the effect of each factor on kinetic parameters in the fermentation of habanero pepper.

| Kinetic Parameter | A: Strain | B: Proportion Puree–Water | C: Maturity | A*B | A*C | B*C | A*B*C |
|---|---|---|---|---|---|---|---|
| $\mu$ | 0.0960 | 0.3085 | 0.0004 * | 0.0770 | 0.2990 | 0.3807 | 0.3130 |
| $\Delta X$ | 0.1793 | 0.0044 * | 0.0012 * | 0.2270 | 0.9640 | 0.0889 | 0.2080 |
| $Q_p$ | 0.1112 | 0.0686 | 0.0183 * | 0.0840 | 0.0451 * | 0.6877 | 0.2965 |
| $\Delta P$ | 0.0072 * | 0.0010 * | 0.0095 * | 0.7427 | 0.1314 | 0.4559 | 0.6279 |
| $P_{max}$ | 0.0027 * | 0.0086 * | 0.0002 * | 0.0852 | 0.4095 | 0.0081 * | 0.3519 |

Note. *p* values < 0.05 mean statistically significant differences and are indicated with an asterisk (*). A*B represents the interaction between factors A (strain) and B (proportion HPP–water). A*C represents the interaction between factors A and C (maturity). B*C represents the interaction between factors B and C. A*B*C stands for the interaction between factors A, B, and C. Abbreviations: $\mu$, growth rate; $\Delta X$, maximum biomass production, $Q_p$, lactic acid production rate; $\Delta P$, maximum lactic acid production; $P_{max}$, maximum productivity.

### 3.3. Evaluation of Volatile Compound Production

According to Table 4, the 2,3-butanedione presented the highest concentrations of all the evaluated volatile compounds, and it was mostly found in the ripe HPP medium with WIL. Limonene and isoamyl isobutyrate were found to have the highest concentrations with COM in the same conditions. Trans-2-hexenal production was mostly the same throughout all fermentation conditions; however, fermentation in unripe 40% medium with WIL had its mean value above the rest. Furthermore, 1-hexanol was mostly found on COM fermentations, with a concentration peaked in unripe 60% medium. In addition, 3,3 dimethyl-1-hexanol and linalool presented slight changes in all of the evaluated fermentations, while cis-3-hexenyl hexanoate was produced mostly in ripe puree medium with COM. Some examples of volatile differences between strains may be observed in their chromatograms (Figures A3 and A4, Appendix B).

The production of volatile compounds varied between each factor, with different effects depending on the volatile compound studied (Table 5). The strain had a significant effect on most volatiles (4: 2, 3 butanedione, limonene, 1-hexanol, and cis-3-hexenyl hexanoate), followed by the proportion of pepper puree–water and maturity (2: limonene and cis-3-hexenyl hexanoate). Interactions were found as well; maturity was the factor with the most interactions (8). Three factors interacted in the production of limonene and cis-3-hexenyl hexanoate. The volatile compounds that were not affected by a specific factor were isoamyl isobutyrate, trans-2-hexen-1-al, and 3,3-dimethyl-1-hexanol.

**Table 4.** Concentrations of volatiles in the fermentation of habanero pepper.

| Volatile Concentration (μg/mL) * | Unripe | | | | Ripe | | | |
|---|---|---|---|---|---|---|---|---|
| | Proportion Puree–Water (% *w/v*) | | | | | | | |
| | 40 | | 60 | | 40 | | 60 | |
| | Strain | | | | | | | |
| | COM | WIL | COM | WIL | COM | WIL | COM | WIL |
| 2,3 Butanedione | 864.96 ± 270.98 [c] | 681.63 ± 22.74 [c] | 326.05 ± 13.87 [a] | 223.95 ± 11.73 [a] | 158.36 ± 34.50 [a] | 579.00 ± 86.15 [b] | 150.37 ± 132.17 [a] | 1324.10 ± 421.22 [d] |
| Limonene | 100.83 ± 0.50 [a] | 144.11 ± 13.74 [b] | 100.60 ± 0.14 [a] | 100.76 ± 0.85 [a] | 101.88 ± 0.64 [a] | Nd | 203.81 ± 5.54 [c] | 100.63 ± 0.21 [a] |
| Isoamyl isobutyrate | 5.54 ± 0.18 [a] | 27.33 ± 6.97 [a] | 6.17 ± 0.21 [a] | 6.81 ± 1.67 [a] | 7.17 ± 3.10 [a] | Nd | 47.62 ± 45.64 [a] | 4.85 ± 0.50 [a] |
| Trans-2-hexen-1-al | 80.19 ± 1.61 [a] | 145.59 ± 35.81 [a] | 82.92 ± 0.67 [a] | 74.73 ± 0.45 [a] | 107.75 ± 9.99 [a] | 90.18 ± 14.87 [a] | 134.27 ± 66.03 [a] | 84.81 ± 4.25 [a] |
| 1-Hexanol | 194.58 ± 39.58 [a] | 182.08 ± 4.92 [a] | 311.63 ± 5.41 [b] | 162.95 ± 2.02 [a] | 243.03 ± 15.03 [b] | 175.48 ± 1.56 [a] | 290.69 ± 89.64 [b] | 185.19 ± 32.05 [a] |
| Hexyl-3-methyl-butanoate | 86.75 ± 1.14 [b] | 83.10 ± 1.44 [b] | 94.83 ± 1.37 [c] | 73.55 ± 0.19 [a] | 75.80 ± 1.35 [a] | 92.08 ± 5.40 [c] | 77.89 ± 4.38 [a] | 79.36 ± 8.87 [a] |
| 3,3 Dimethyl-1-hexanol | 125.43 ± 8.53 [a] | 129.04 ± 17.36 [a] | 134.05 ± 3.03 [a] | 118.30 ± 0.61 [a] | 140.34 ± 24.71 [a] | 128.54 ± 14.69 [a] | 153.78 ± 12.43 [a] | 128.67 ± 7.22 [a] |
| Linalool | 137.89 ± 0.01 [a] | 137.92 ± 0.02 [a] | 137.95 ± 0.11 [a] | 138.01 ± 0.06 [b] | 138.11 ± 0.02 [b] | 137.92 ± 0.03 [a] | 137.93 ± 0.11 [a] | 137.91 ± 0.08 [a] |
| cis-3- Hexenyl hexanoate | 89.09 ± 1.46 [a] | 95.83 ± 2.26 [b] | 92.84 ± 1.56 [b] | 85.37 ± 0.48 [a] | 141.68 ± 9.02 [d] | 86.76 ± 1.74 [a] | 113.75 ± 4.83 [c] | 84.08 ± 1.34 [a] |

Note. * Concentrations of all volatile compounds except for 2,3 butanedione are multiplied by $10^2$. Data are expressed as means. Values in the same row that do not share a lower-case letter are statistically different. Abbreviations: COM, commercial *L. plantarum*; WIL, wild *L. plantarum*; Nd, not detected.

**Table 5.** *p* values for the effect of each factor on volatile compound concentrations in the fermentation of habanero pepper.

| Volatile Compound | A: Strain | B: Proportion | C: Maturity | A*B | A*C | B*C | A*B*C |
|---|---|---|---|---|---|---|---|
| 2,3 Butanedione | 0.0237 * | 0.9704 | 0.5069 | 0.3611 | 0.0232 * | 0.0345 * | 0.1335 |
| Limonene | 0.0000 * | 0.0000 * | 0.0052 * | 0.0029 * | 0.0000 * | 0.0000 * | 0.0041 * |
| Isoamyl isobutyrate | 0.4254 | 0.4599 | 0.6848 | 0.1214 | 0.0581 | 0.0817 | 0.6707 |
| Trans-2-hexen-1-al | 0.8619 | 0.4152 | 0.5563 | 0.0900 | 0.0528 | 0.1413 | 0.4678 |
| 1-Hexanol | 0.0019 * | 0.0687 | 0.5752 | 0.0462 * | 0.8764 | 0.5982 | 0.2203 |
| Hexyl-3-methyl-butanoate | 0.4058 | 0.1776 | 0.1487 | 0.0042 * | 0.0008 * | 0.2960 | 0.7399 |
| 3,3 Dimethyl-1-hexanol | 0.1025 | 0.6788 | 0.1331 | 0.2546 | 0.3794 | 0.5722 | 0.8258 |
| Linalool | 0.4215 | 0.8585 | 0.5038 | 0.1519 | 0.0548 | 0.0309 * | 0.3166 |
| cis-3- Hexenyl hexanoate | 0.0000 * | 0.0013 * | 0.0000 * | 0.1910 | 0.0000 * | 0.0148 * | 0.0009 * |

Note. *p* values < 0.05 mean statistically significant differences and are indicated with an asterisk (*). A*B represents the interaction between factors A (strain) and B (proportion). A*C represents the interaction between factors A and C (maturity). B*C represents the interaction between factors B and C. A*B*C stands for the interaction between factors A, B, and C.

A principal component analysis (PCA) was performed, associating the factors of maturity (Figure 2a) and strain (Figure 2c) with the volatile compounds produced; each of them had its correspondent k means chart (Figure 2b,d). As for maturity (Figure 2a), most of the data (99.48%) are represented by the horizontal axis, and the rest (0.52%) are represented by the vertical axis, which is where maturity varies. It seemed that none of the analyzed volatile compounds were characteristic for one maturity stage, since they were mostly close to the center of the chart. This was confirmed by its k means clusters (Figure 2b), which were on both the positive and the negative sides of the vertical axis or very close to 0.0 (isoamyl isobutyrate and 1-hexanol clusters). The strain factor (Figure 2c) varied on the vertical axis as well, with a 3.28% of representation of the data. Another observation made from this chart revealed that 1-hexanol seemed to be associated with COM, while 3,3 dimethyl-1-hexanol and linalool, with WIL. This was validated by their k means clusters, which aligned to those of their respective variables.

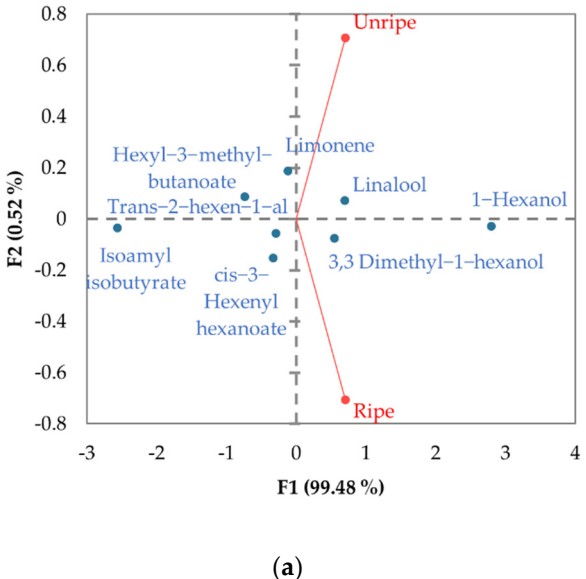

(a)

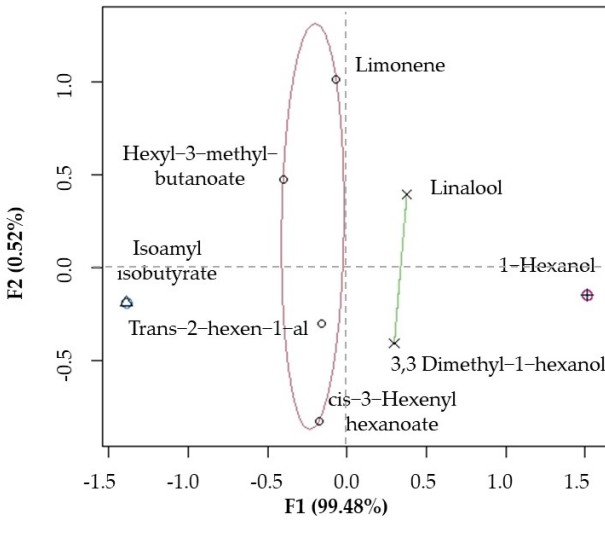

(b)

**Figure 2.** *Cont*.

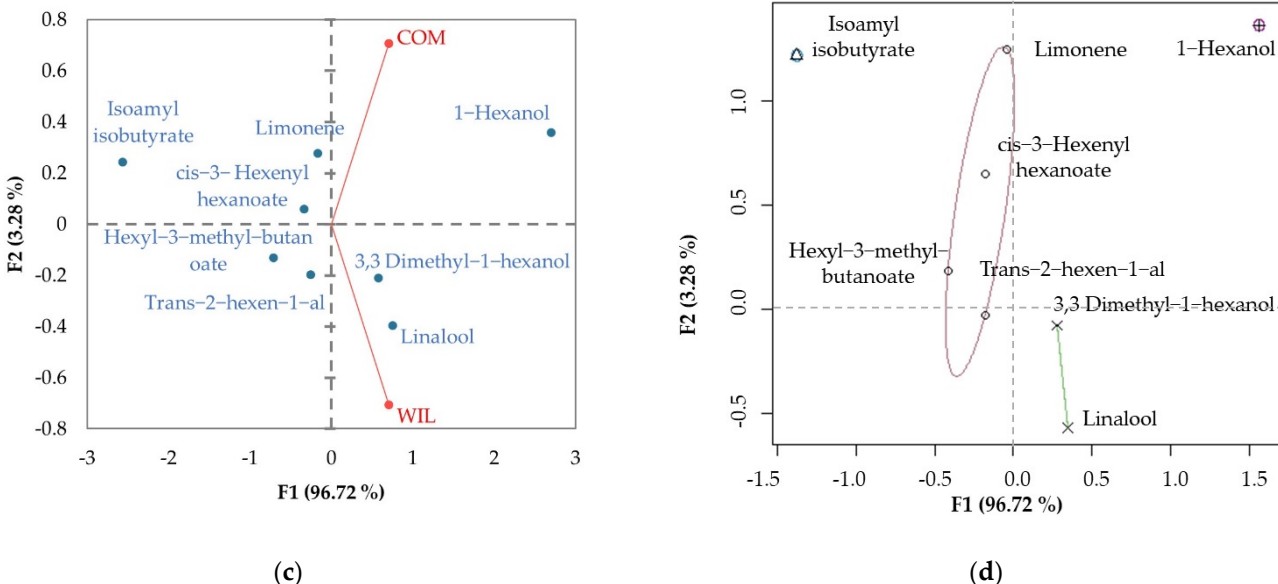

(**c**)　　　　　　　　　　　　　　　　(**d**)

**Figure 2.** Analysis of volatile compounds production: Principal component analysis (PCA) depending on (**a**) maturity and (**c**) strain used; and clusters of k means depending on (**b**) maturity and (**d**) strain used. Abbreviations: COM, commercial *L. plantarum*; WIL, wild *L. plantarum*.

## 4. Discussion

As seen in Table 3, kinetic parameters were affected mostly by maturity, followed by the proportion of habanero pepper puree–water, both of which have the availability of carbohydrates in common. Ripe habanero pepper puree was expected to have a higher free sugar concentration than the unripe sample, as seen with other species of *Capsicum* [18], due to the enzymatic degradation of complex carbohydrates [19]. Furthermore, at the time when the metabolic process in *L. plantarum* is focused on cell division, it metabolizes carbohydrates via the Embden–Meyerhof pathway, the product of which is mostly lactic acid [20]. Prior studies have reported that *L. plantarum* presents better biomass and lactic acid production (with higher production and higher rates) with mediums supplemented by mono-and disaccharides such as glucose and lactose rather than those supplemented by complex oligosaccharides [21]. This might be the reason why in lactic acid production (Figure 1b), both strains had the same behavior in a 40% ripe HPP medium as those in a 60% unripe HPP medium because they might have a similar free sugar concentration. Regarding the effect of the strains on the kinetic parameters, they only showed differences in lactic acid production. This might be attributed to the source of isolation for each strain (COM was isolated from *Solanum lycopersicum* as declared by the manufacturer, while WIL, from habanero pepper) since both bacteria are from the same species; for example, *L. plantarum* that are isolated from fecal matter do not metabolize xylose or raffinose, which is common in *L. plantarum* strains that ferment vegetables [22]. WIL was more adapted to the habanero pepper medium due to its origin, thus metabolizing carbohydrates more efficiently and producing more lactic acid at a faster rate. COM was isolated from another vegetable source, which means it may not be able to transform some complex carbohydrates present in habanero pepper, as it is not used for this purpose, developing lower lactic acid production. Some studies are uncertain of the effect of the source of isolation on the differences among *L. plantarum* strains, assuming their variations occur randomly, a condition that, in fact, occurs with strains of *L. paracasei* [6]. However, the source of isolation may have an effect according to the results found.

As regards the volatile compounds, differences were found to depend on every factor. One interesting finding is that 2,3 butanedione production was mostly affected by the strain, having the highest concentration on WIL in the 60%, ripe HPP medium, which is the one with the highest concentration of free sugars expected. This compound is related to the

exogenous pyruvate catabolism of *L. plantarum*, as it is directly derived from acetoin and linked to the late stages of the fermentation, as acetoin is mostly produced when glucose is at a low concentration, and pH is at its minimum [21]. As expected, concentrations of limonene and linalool did not vary after fermentation, as they are common terpenes in habanero pepper. Other compounds whose concentrations remained unchanged were isoamyl isobutyrate, trans-2-hexen-1-al, and 3,3 dimethyl-1-hexanol [12–14]. Limonene was affected by every factor; this might be due to a process where limonene is obtained by other modified terpenes depending on the energetic needs of the culture [8]. The low concentrations of linalool (traces) resulted in variations in the proportion of HPP–water and maturity. The esters hexyl-3-methyl-butanoate and cis-3-hexenyl hexanoate were mostly affected by maturity and the strain. Concentrations of these compounds have been reported to change through ripeness; moreover, they are also involved in the metabolism of amino acids and fatty acids [12–14]. Through conducting a principal component analysis, it was observed that 1-hexanol was mostly associated with COM; however, it might detrimental, as it is associated with off-flavors in other vegetables such as mung beans, in which hexanol and hexanal are assumed to be reduced through fermentation. Further, they are transformed into esters with a more pleasant fragrance through acid dehydrogenase activity; thus, WIL might be a better choice for the fermentation of habanero pepper due to the reduction of 1-hexanol through fermentation [23]; the cluster including linalool and 3,3 dimethyl-1-hexanol was associated with WIL; however, closeness to x-axis showed a slight relation. In addition, neither compound was specifically associated with maturity, even when ripe habanero was related to esters, alcohols, and terpenes, and unripe habanero was related to aldehydes. It should also be noted that volatile concentration might not be the only factor affecting the analysis, as it is not always related to its aroma impact, so differences between compounds with lower concentrations might be observed [12–14].

**5. Conclusions**

The best conditions to produce both biomass and lactic acid were the use of 60% ripe HPP with WIL. Differences in growth mostly depended on the medium, with the proportion of 60% HPP–40% water on a ripe maturity as the one with the highest μ and ΔX. However, in lactic acid production, both strains differed—WIL, with 60% ripe HPP, was the most favorable for $Q_p$, ΔP, and $P_{max}$ values.

The volatile composition was mostly affected by the strains used. WIL had an edge in the production of 2,3 butanedione, while COM exceeded in the production of limonene, 1-hexanol, and cis-3-hexenyl hexanoate. Cis-3-hexenyl hexanoate and limonene were affected by almost all the analyzed factors; COM presented the highest production of the first in a 40% ripe HPP medium and the highest production of the latter in a 60% ripe HPP medium. Finally, through a PCA analysis, volatile compounds were clustered according to the maturity of the pepper for which no associations were found. This may be due to other factors affecting the volatile compositions, such as the concentration of amino and fatty acids, or the aroma impact since there are sensory impactful compounds such as limonene from peppers, which are normally found in trace concentrations. However, PCA showed that 1-hexanol could be associated with COM, likely through the metabolism of fatty and amino acids, while a specific volatile compound for WIL from the volatiles analyzed was not found. These findings could be applied in the food industry to obtain fermented products based on habanero pepper, with organoleptic characteristics that are prominent among similar products on the market. In addition, added value will be provided to the final product, improving the commercial and cultural importance of the habanero pepper in the Yucatan peninsula.

**Author Contributions:** This study was made possible through the collaboration of all authors. Conceptualization, I.M.R.-B. and D.L.-S.; methodology, J.E.O.-M. and D.L.-S.; software E.R.-R. and D.L.-S.; validation, I.M.R.-B.; formal analysis I.M.R.-B. and M.O.R.-S.; investigation I.M.R.-B. and D.L.-S.; resources, I.M.R.-B.; data curation, I.M.R.-B., J.E.O.-M. and D.L.-S.; writing—original draft preparation, D.L.-S. and J.E.O.-M.; writing—review and editing, I.M.R.-B. and M.O.R.-S.; visualization,

D.L.-S.; supervision, I.M.R.-B., project administration, I.M.R.-B.; funding acquisition, I.M.R.-B. All authors have read and agreed to the published version of the manuscript.

**Funding:** This research received no external funding.

**Institutional Review Board Statement:** Not applicable.

**Informed Consent Statement:** Not applicable.

**Data Availability Statement:** All the data are available in the manuscript file.

**Acknowledgments:** The authors would like to thank the National Council of Science and Technology of Mexico (CONACYT) for providing scholarship numbers 715424 and 1051122 for J.E.O.M. and D.L.S., respectively.

**Conflicts of Interest:** The authors declare no conflict of interest.

## Appendix A

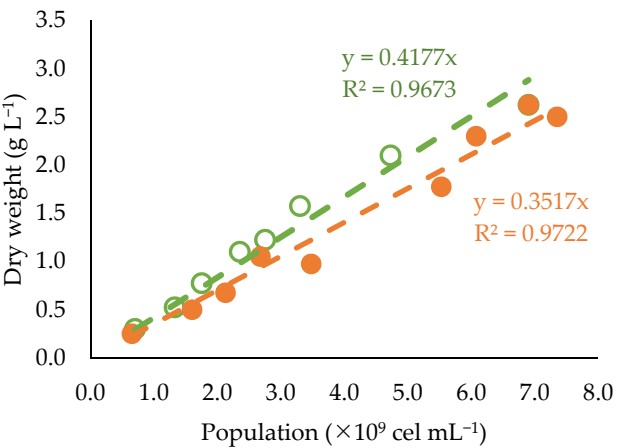

**Figure A1.** Correlation curve between dry weight and population for each strain. Legend: ● COM ○ WIL. Abbreviations: COM, commercial *L. plantarum*; WIL, wild *L. plantarum*.

## Appendix B

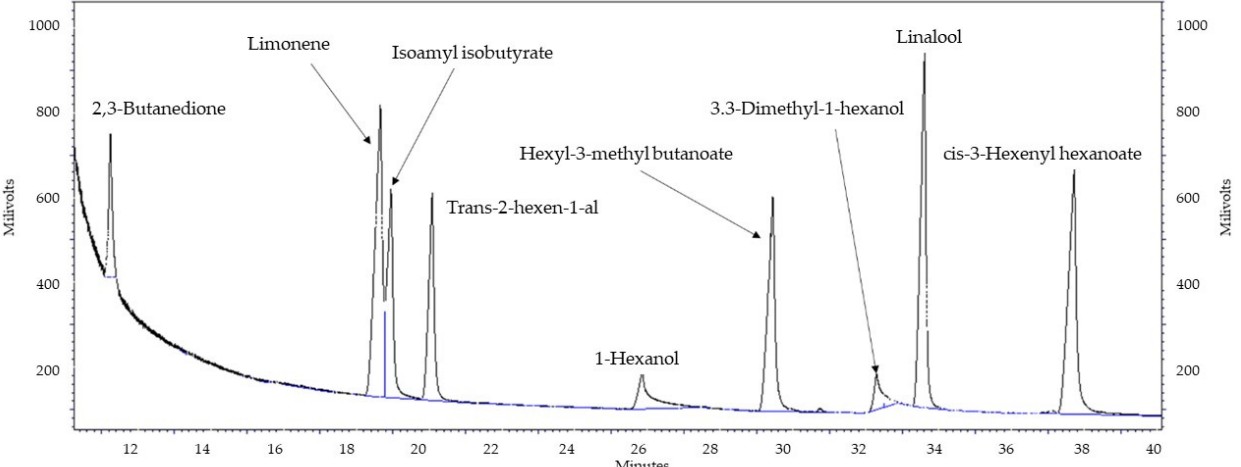

**Figure A2.** Chromatogram for volatile standards.

**Table A1.** Retention time and adjustment quality for the calibration curves of volatile standards.

| Volatile Compound | Retention Time (min) | $R^2$ |
|---|---|---|
| 2,3 butadione | 11.27 | 0.9933 |
| Limonene | 18.78 | 0.9920 |
| Isoamyl isobutyrate | 19.10 | 0.9930 |
| Trans-2-hexen-1-al | 20.18 | 0.9924 |
| 1-Hexanol | 25.94 | 0.9925 |
| Hexyl-3-methyl butanoate | 29.57 | 0.9923 |
| 3,3-Dimethyl-1-hexanol | 32.36 | 0.9951 |
| Linalool | 33.70 | 0.9965 |
| Cis-3-Hexenyl hexanoate | 37.86 | 0.9912 |

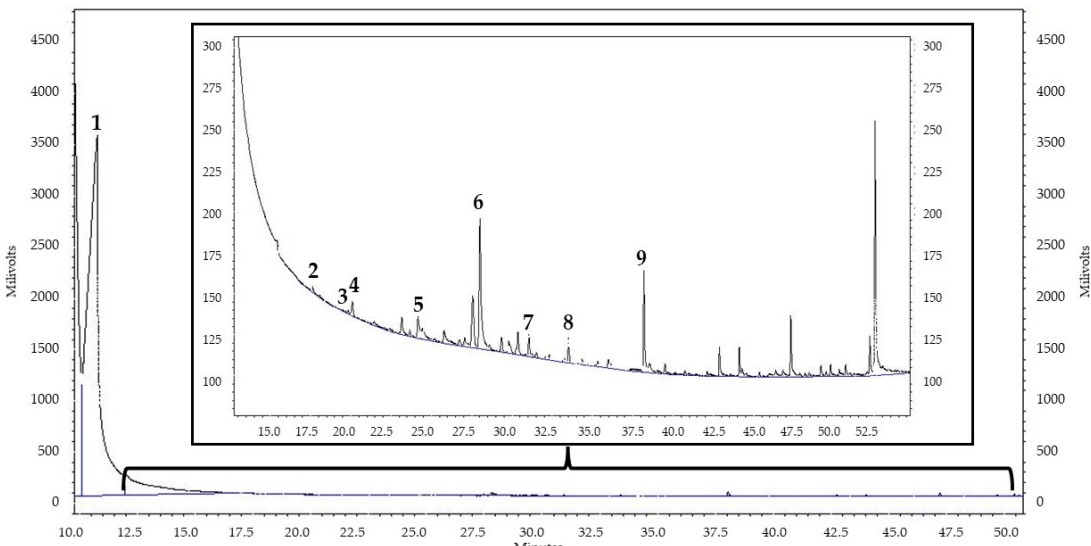

**Figure A3.** Chromatogram for a WIL fermented sample in a 60% ripe habanero pepper, 40% water medium. Legend: 1, 2,3 butanedione; 2, limonene; 3, isoamyl isobutyrate; 4, trans-2-hexen-1-al; 5, 1-hexanol; 6, hexyl-3-methyl butanoate; 7, 3,3-dimethyl-1-hexanol; 8, linalool; 9, cis-3-hexenyl hexanoate.

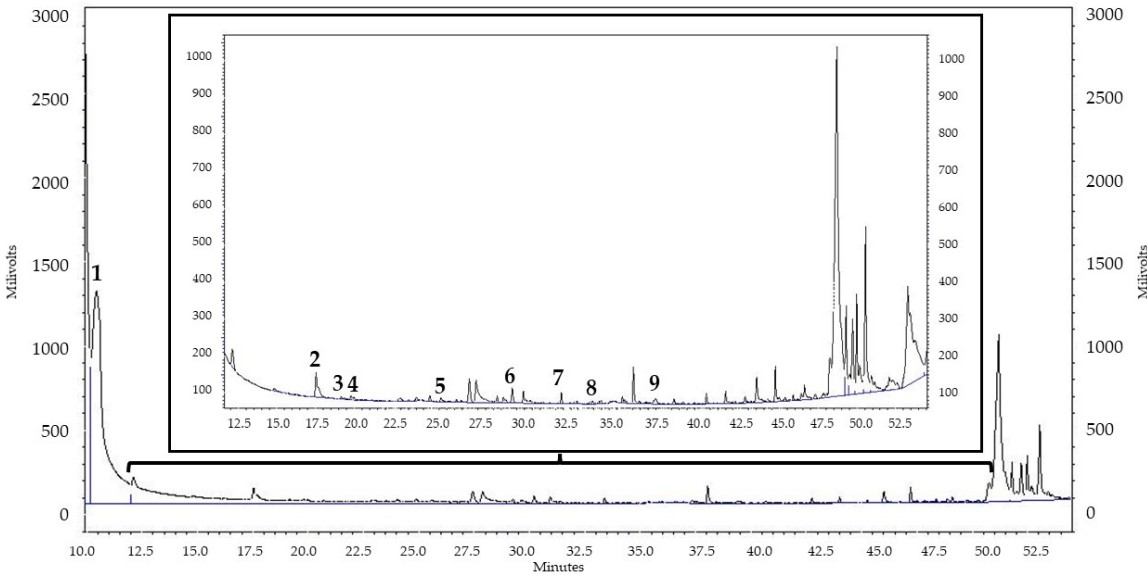

**Figure A4.** Chromatogram for a COM fermented sample in a 60% ripe habanero pepper, 40% water medium. Legend: 1, 2,3 butanedione; 2, limonene; 3, isoamyl isobutyrate; 4, trans-2-hexen-1-al; 5, 1-hexanol; 6, hexyl-3-methyl butanoate; 7, 3,3-dimethyl-1-hexanol; 8, linalool; 9, cis-3-hexenyl hexanoate.

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
