# Peer review of "Fermentation of Habanero Pepper by Two Lactic Acid Bacteria and Its Effect on the Production of Volatile Compounds"

_fermentation, doi:10.3390/fermentation8050219_

Round 1

Reviewer 1 Report

The authors have studied the fermentation of habanero peppers using two Lactiplantibacillus plantarum strains (one isolated from habnaero peppers and the other a commercial starter), and the stage of ripeness of habnero peppers as variables. The production of bacterial biomass, lactic acid, and voltatiles were studied. The overall conclusion was that the L. plantarum strain obtained from habanero pepper was better adapted to grow in the pepper-puree nad also producing more the flavour component 2,3 butanedione, while no other clear marked differences between the strains were detected (except for the association of 1-hexanol with te commercial L. plantarum strain) , the main factor affecring the outcome of the fermentation being the stage of maturity of the peppers.

The resarch appears comtetently done, and I do not have comments related to the outcome of the experiments. There are, however, certain points that should be addressed.

1) The taxoomy and nomenclature of the former genus Lactobacillus has been thoroughly revised lately, and the authors should take this into account.

2) Because two strains of L. plantarum were compared, it would be of interest to check if there are any differences in their carbohydrate fermentation spectra. This could be easily done by, for example a simple API test

3) Was any organoleptic assessment of the fermented habnero peppers considerd?

Reviewer 2 Report

The topic of the article falls within the thematic scope of the journal FERMENTATION.

The objective of study was to compare two different L. plantarum strains (one wild - isolated from Yucatan habanero pepper and a commercial strain) on the basis of the kinetic parameters of fermentation and the composition of volatile compounds produced.

I have no objections to the methods used and the general presentation of the results and their discussion.

I have only a few minor comments and a more important note regarding the lack of a sufficient description of the method (section 2.4.2).

All comments and suggestions for corrections were introduced in the review mode to the attached pdf file.

Reviewer 3 Report

Dear Authors,

I appreciate your work on the manuscript which aimed to compare habanero puree fermentation process by two L. plantarum strains. Ensiling via lactic acid bacteria fermentation is a common but still very important method for food and feed preservation. Moreover, the microbial process provides many advantages like changes in metabolites content.

I recommend your article for publication and I would like to suggest some minor changes:

  • abstract: please add if the production of more than 5000 tons reflects to one year, globally or per country? Please describe why did you choose two strains of L. plantarum in the study. Explain how the origin of isolation determines some strains abilities;
  • table 2 - please describe the abbreviations in the table, you mention them in materials and methods but it will be easier to analyze the table if you write the abbreviation below table once again;
  • new name for Lactobacillus plantarum species should be used
